# Supercritical Fluid Extraction of Phenolic Compounds from Mango (*Mangifera indica* L.) Seed Kernels and Their Application as an Antioxidant in an Edible Oil

**DOI:** 10.3390/molecules26247516

**Published:** 2021-12-11

**Authors:** Luis Miguel Buelvas-Puello, Gabriela Franco-Arnedo, Hugo A. Martínez-Correa, Diego Ballesteros-Vivas, Andrea del Pilar Sánchez-Camargo, Diego Miranda-Lasprilla, Carlos-Eduardo Narváez-Cuenca, Fabián Parada-Alfonso

**Affiliations:** 1Food Chemistry Research Group, Departamento de Química, Facultad de Ciencias, Universidad Nacional de Colombia, Sede Bogotá, Carrera 45 No 26-85, Bogotá 111321, Colombia; lmbuelvasp@unal.edu.co (L.M.B.-P.); gafrancoar@unal.edu.co (G.F.-A.); cenarvaezc@unal.edu.co (C.-E.N.-C.); 2Departamento de Ingeniería, Universidad Nacional de Colombia, Sede Palmira, Carrera 32 No. 12-00, Palmira 763531, Colombia; hamartinezco@unal.edu.co; 3Departamento de Nutrición y Bioquímica, Facultad de Ciencias, Pontificia Universidad Javeriana, Bogotá 111321, Colombia; diego_ballesteros@javeriana.edu.co; 4Grupo de Diseño de Productos y Procesos (GDPP), Department of Food and Chemical Engineering, Universidad de los Andes, Carrera 1E No. 19 A 40, Edificio Mario Laserna, Bogotá 111711, Colombia; ad.sanchez@uniandes.edu.co; 5Facultad de Ciencias Agrarias, Universidad Nacional de Colombia, Sede Bogotá, Carrera 45 No 26-85, Bogotá 111321, Colombia; dmirandal@unal.edu.co

**Keywords:** mango seed kernel, sunflower edible oil, oxidative stability, Rancimat method, phenolic compounds, supercritical fluid extraction

## Abstract

Phenolic compounds from mango (*M. indica*) seed kernels (MSK) *var*. Sugar were obtained using supercritical CO_2_ and EtOH as an extraction solvent. For this purpose, a central composite design was carried out to evaluate the effect of extraction pressure (11–21 MPa), temperature (40–60 °C), and co-solvent contribution (5–15% *w*/*w* EtOH) on (i) extraction yield, (ii) oxidative stability (OS) of sunflower edible oil (SEO) with added extract using the Rancimat method, (iii) total phenolics content, (iv) total flavonoids content, and (v) DPPH radical assay. The most influential variable of the supercritical fluid extraction (SFE) process was the concentration of the co-solvent. The best OS of SEO was reached with the extract obtained at 21.0 MPa, 60 °C and 15% EtOH. Under these conditions, the extract increased the OS of SEO by up to 6.1 ± 0.2 h (OS of SEO without antioxidant, Control, was 3.5 h). The composition of the extract influenced the oxidative stability of the sunflower edible oil. By SFE it was possible to obtain extracts from mango seed kernels (MSK) *var*. Sugar that transfer OS to the SEO. These promissory extracts could be applied to foods and other products.

## 1. Introduction

Mango (*Mangifera indica* L.) is one of the most consumed fruits worldwide [1,2]. In Colombia, mango is among the most important permanent fruit crop, with a production of 325,000 t in 2019 [3]. More than 20 different cultivars have been described [4], with mango *variety* Sugar and *var*. Tommy Atkins as the most important ones for the Colombian agro-industry [5]. In fruit processing, the main by-products generated are epicarp (peel) and seed. These by-products represent around 35–60% of the fruit’s total weight. The seed can reach up to 75,000 t/year, becoming a source of contamination without any sustainable disposal at present. This agricultural by-product, especially the mango seed kernels (MSK), is considered, however, an important source of biomolecules that could be exploited [1,6,7,8,9,10].

Nowadays, almost all processed foods include synthetic antioxidants. For instance, in the oil industry, butylated hydroxytoluene (BHT), butylated hydroxyanisole (BHA) or tert-butylhydroquinone (TBHQ) are the most commonly used [11,12]. Negative health effects, however, have been reported due to the consumption of synthetic antioxidants, which could be avoided by using natural antioxidants instead [13,14,15] such as those present in MSK. Recent work has described the presence of different families of phenolic compounds with interesting antioxidant properties [6]. Antioxidants extracted from natural sources might, therefore, be a good alternative to synthetic compounds.

Alternative extraction methods, such as supercritical fluid extraction (SFE), were explored to obtain enriched-biomolecules extracts with antioxidant properties. The use of supercritical CO_2_ as an extraction solvent reduces not only the consumption of liquid solvents but also evaporation steps because it is gaseous under ambient conditions. In addition, it provides an inert environment during the extraction process that helps to prevent the degradation of compounds sensitive to oxidation and allows high selectivity [16,17].

On the basis of their chemical structure, the phenolic compounds found in MSK are of considerable polarity, which makes their extraction with supercritical CO_2_ difficult. To increase the polarity of the extraction system small amounts of polar co-solvents, such as ethanol (EtOH), are used in combination with supercritical CO_2_ [18,19,20,21,22]. Our previous works from guava seed studies have found that SFE using supercritical CO_2_ with cosolvent (ethyl acetate or ethanol) led to better results in yield and extract quality (antioxidant activity), EtOH being the best co-solvent [23]. As a result an increase in the overall extraction yield and the compounds extraction efficiency, including mangiferin, gallic acid, and ellagic acid, present in mango fruit, might be expected [1,6,7,10,24]. To the best of our knowledge, the use of supercritical CO_2_ with EtOH as co-cosolvent has not been used to extract biomolecules from MSK. Additionally, while the action mechanism of pure phenolic compounds on bulk oil and oil in water emulsion systems is well understood [25], studies of crude extracts are less well documented [12,26].

The main aim of this research was optimizing the production of an extract from MSK, using supercritical CO_2_ and EtOH as co-solvent the best extract would transfer the major oxidative stability (OS) to sunflower edible oil (SEO). The supplementary aims of this research were (i) to evaluate the total phenolics content (TPC), the total flavonoids content (TFC), and the in vitro antioxidant activity by the DPPH method on such extracts, (ii) to characterize the extracts by means of reversed-phase (ultra high-performance liquid chromatography with diode array detector (RP-UHPLC-DAD)), and (iii) to evaluate the relationship between the TPC, TFC, DPPH, and composition of the extracts, and their antioxidant activity on a bulk sunflower oil system.

The OS was studied using the Rancimat method due to its high performance to evaluate the OS of the fats and oils, as well as, to assay the antioxidant activity of extracts and compounds [27]. In the same way, the DPPH method was selected to measure antioxidant activity because it was the most used method for the in vitro antioxidant activity evaluation [28]. This work proposes an application for important agro-industrial byproducts, the MSK var. sugar, which could be used as a source of antioxidant extract applied in the fats and oils industry, a topic which is framed in the guidelines of green chemistry.

## 2. Results and Discussion

### 2.1. Extraction Yield

The yield results (soluble fraction-*SF* yield, unsoluble fraction-*uSF* yield, and global extraction yield, *SF* + *uSF*) obtained under the SFE and the Soxhlet methods are shown in Figure 1. The highest overall and *SF* yields (22.5 and 7.7%, respectively), using the SFE technology, were obtained at 11.0 MPa, 60.0 °C and 15.0% EtOH. Under the Soxhlet (SX) method the overall yield reached 18.9% and the *SF* yield 7.1%. The maximum extraction yield for *uSF* under SFE (14.7%) was similar to those obtained by Jahurul et al. [29], which reported values of up to 13.5% of fat extracted by supercritical CO_2_ from the kernel of six mango varieties from Malaysia [30]. Our research shows that under the SX method the overall extraction yield was higher than those reported for the *var.* Kibangou (13.0%) [31] and the *var.* Kent (12.3%) [32], both extracted with petroleum ether from the mango seed kernel (MSK). Additionally, a research work using the SX extraction method and involving the kernel of 20 mango Colombian varieties reported a maximum overall extraction yield of 11.8% for the *var.* Rosa, using petroleum ether as solvent [33]. Furthermore, the highest overall extraction yield as obtained in the current research under SFE (22.5%) is far better than all those previously reported, using the SX extraction method. The higher extraction yield under SFE relative to that under Soxhlet may be due to the properties offered by the SFE technique. Changes in pressure and co-solvent cause an increase in density and polarity which is translated into an increase in the solubility of compounds, because of greater interaction between the extractant agent and the matrix [20]. Similarly, an increase in extraction temperature decreases the viscosity and facilitates the diffusion of compounds, thus allowing rapid exhaustion of the sample and increasing the yield.

### 2.2. Oxidative Stability of Sunflower Edible Oil with Added Extracts

Figure 2 shows the results for sunflower edible oil (SEO) having added the extracts obtained under SFE and SX. It also shows the results for the OS of SEO with TBHQ, and those of the control without antioxidants. Most of the extracts (SFE and SX) increased the t_i_ with respect to the control (3.5 h), with TBHQ exhibiting the greatest t_i_ (7.8 h). The extract yielding the greatest t_i_ was obtained at 21.0 MPa, 60.0 °C, 15.0% EtOH (6.1 h), its protection factor (PF) was 1.7, equivalent to 78% of the effect generated by the TBHQ (PF 2.2).

To the best of our knowledge, there are no reports on the use of extracts obtained from MSK to protect edible oils from oxidation. Our results, however, can be compared with extracts from other biomasses. For instance, Asnaashari et al. [34] obtained extracts from *Rubus fruticosus* by conventional extraction using methanol and evaluated their antioxidant effect in SEO at 1000 mg Kg^–1^, 120 °C, and 20 L h^–1^. Under such conditions, the t_i_ and PF reached values of 10.8 h and 2.2, respectively. Promisingly, those values were higher than those obtained with 200 mg BHT Kg^–1^ (9.8 h and PF 1.9). Another research, conducted by Upadhyay et al. [35], evaluated the effect of extracts obtained from *Rosmarinus officinalis*. They tested such extracts under the same Rancimat method conditions of this research and obtained a similar PF (1.6) to that given by the current research.

### 2.3. Experimental Design on the Supercritical Fluid Extraction

The experimental conditions (P, T, and %EtOH) are shown in Table 1.

The chosen variable main effects on the ability of the obtained extracts to increase the OS of the SEO as measured by the t_i_ are shown in Figure 3a. Although most of the variables together with their interactions had a statistically significant effect, it was observed that the variable that positively influenced the oxidative stability the most was the co-solvent factor, followed by pressure. Because EtOH is a polar solvent, the increase in its concentration in the extractant phase generates greater polarity in this phase, favoring the extraction of polar analytes, thus showing the modifying effect of the co-solvent.

The coefficients of the mathematical model that better described the t_i_ as a function of pressure, temperature, and co-solvent are shown in Table 2. Under such circumstances, the adjusted R-squared statistic indicates that the model explains 92.4% of the variability in the t_i_. The lack-of-fit test (*p* = 0.06) was also indicative of the adequacy of the mathematical model. According to the coefficients presented in Table 2, the optimum conditions predicted in the explored region correspond to 24.4 MPa, 67.0 °C and 18.4% EtOH, which was similar to the extract obtained at 21.0 MPa-60.0 °C-15.0% EtOH (See Table 1). Figure 3b shows the surface response obtained from the predicted induction time according to the mathematical model (Table 2). Such surface shows that increasing pressure and co-solvent percentage under SFE increases the induction time.

To explain how the composition of the extracts responds to the observed t_i_, the TPC, TFC, DPPH, and individual phenolics in the extracts were studied and their contribution to the t_i_ was analyzed by mathematical models.

### 2.4. Total Phenolics and Total Flavonoids Contents, and DPPH Radical Antioxidant Activity

The TPC, TFC, and DPPH in the *SF* obtained by SFE as well as in the SX extraction are shown in Figure 4a–c.

The greatest value for the TPC was obtained when extraction was made under the SX method (62.1 mg-eq GA g^−1^ extract). The highest TPC value under SFE was obtained at 11.0 MPa, 60.0 °C, and 15.0% EtOH (57.3 mg-eq GA g^−1^ extract). Under these SFE conditions, the TFC was also the highest (13.6 mg-eq Q g^−1^ extract). The higher TPC under the SX extraction (relative to SFE) can be related to its high solvation power, typical of liquid solvents, and its high polarity, which is higher than that of supercritical CO_2_-EtOH [36]. This fact can also be observed taking into account the Hildebrand solubility parameters, where values of 26.2, 3.5, and 6.3 are reported for EtOH, supercritical CO_2_, and supercritical CO_2_-EtOH, respectively [37,38]. The results given by the current research were lower than those reported by Khammuang et al. for the kernel of mango *var.* Thai, which reported values of TPC and TFC of 118.1 mg-eq AG g^−1^ extract and 110.1 mg-eq Q g^−1^ extract, respectively, when the extraction was made by solid–liquid extraction using water [39]. Our results, however, were greater than those reported by Ballesteros for an SX methanolic extract of MSK *var.* Sugar with 16.2 mg-eq GA g^−1^ extract and 2.0 mg-eq Q g^−1^ extract [40]. TFC to TPC proportions obtained from MSK by SFE were 1:5 approximately, these values contrast with the same proportions obtained from MSK by PLE (they were 1:100 approximately), these results are in agreement with the high selectivity of SFE [8], PLE’s obtained extract yields were better but lower selectivity than those obtained by SFE. On the other way, the differences observed between our results and those from elsewhere could be due to differences in the agroclimatic conditions of cultivation and the varieties of mango, among others [12].

Under SFE, when measuring the DPPH radical scavenging activity, the most active extract was obtained at 11.0 MPa, 60.0 °C, and 15.0% EtOH (120.0 µmol-eq Trolox g^–1^ extract), corresponding to 80.6% inhibition of DPPH radical. Meanwhile, the SX extract was less active showing an inhibition of 55.5%, which was equivalent to 54 µmol-eq trolox g^–1^ extract. The best results obtained here were comparable to those obtained by El-Baroty et al., where they achieved similar inhibition activities for methanolic and aqueous extracts (up to 89% inhibition of DPPH radical) for the kernel of the mango *var.* Zebdeia [41,42]. The DPPH antioxidant activity of the most active extract referred to dry sample (0.931 mmol-eq trolox 100 g^–1^ dry sample) was lower, however, than those reported by Dorta et al., who obtained values of 127.8 mmol-eq trolox 100 g^–1^ dry sample for extracts from the kernel of the mango *var.* Keitt, using a microwave-assisted extraction technique [43].

### 2.5. Identification and Quantification of Phenolic Compounds in the Extracts

The analysis of the extract obtained under SFE with the highest t_i_ (experiment #10 in Table 1) gave a chromatogram dominated by ellagic acid (compound **2**), followed by gallic acid (compound **1**), and other unknown compounds (Figure 5). Although compounds **3**–**8** were found in low abundance in experiment #10, these were abundant in other extracts obtained under the SFE and the SX extraction methods (Appendix A). Gallic acid together with ellagic acid were previously reported in the varieties of mango: Keitt, Osteen, Sensation, and Gomera [6,7,40]. The quantification of compounds revealed a variation for gallic acid (compound **1**) contents from 8.4 to 59.9 mg gallic acid g^–1^ extract and for ellagic acid (compound **2**) from 7.1 to 178.0 mg ellagic acid g^–1^ extract. The extract that generated the greatest OS to SEO (SFE experiment 10) content 59.9 mg gallic acid g^–1^ extract and 163.8 mg ellagic acid g^–1^ extract.

### 2.6. Correlations among Variables

Evaluation of Pearson correlation coefficients revealed that TPC, TFC, and DPPH radical scavenging activity were positively correlated among them (*p* < 0.0001). The correlation study between the concentration of each individual phenolic compound with the values of TPC, TFC, and DPPH was made. When doing so, compounds **3**–**7** were positively correlated with TPC (*p*-values ranging from 0.00001 to 0.0020), compounds **1**, **3**–**5**, and **7** were positively correlated with TFC (*p*-values ranging from 0.00001 to 0.0200), and DPPH was positively correlated with compounds **3**–**7** (*p*-values ranging from 0.00001 to 0.0100). The lack of any correlation between gallic acid and DPPH could be due to its low radical scavenging activity [44]. Despite ellagic acid having been shown to exhibit good DPPH antioxidant activity as compared to quercetin, chlorogenic acid or gallic acid [44] when tested in a pure form, no statistical correlation between DPPH and the concentration of ellagic acid was found in the current research. These results suggest that compounds **3**–**7** have stronger radical scavenging activity than either gallic acid or ellagic acid.

### 2.7. Induction Time and the Composition of the Extract

To evaluate if the composition of the extract (obtained under SFE or SX extraction) was responsible for the induction time observed when extracts were added to an SEO, different mathematical approaches were tested. The response variable t_i_ was not correlated to TPC, TFC, and DPPH when these three were evaluated as independent variables in linear or quadratic mathematical models with backward stepwise elimination. In those scenarios, despite the low values for the absolute standard deviation, the mathematical models did not represent well the variation observed in the t_i_ as judged by the low value for the adjusted R^2^. The adjusted R^2^ and absolute standard deviation values reached 33.1 and 8.9% for the linear mathematical model with stepwise backward elimination and 61.0 and 6.6% for the quadratic model with stepwise backward elimination. That is to say that the extracts with the highest TPC, TFC, and DPPH values were not those that protected the SEO from lipid oxidation. Therefore, the reducing power (as measured by the TPC) and radical scavenging activity (accessed by the DPPH method) of the extracts did not explain their behavior in the edible oil.

In a previous work of our research group, a protection effect of a crude extract from *Rubus glaucus* waste against lipidic oxidation in an edible oil emulsion was found [12]. This protecting effect was partially related to the TPC and the partition coefficient. This last parameter is characteristic of each individual phenolic compound. In the current research, a clear relationship was not found between the values of TPC, TFC or DPPH. To understand how a crude extract can protect bulk oil against lipid oxidation, the concentration effect of each individual compound present in the crude extract was studied. On the basis of the coefficients shown in Table 3, mathematical modeling indicated that gallic acid (compound **1**), together with compounds **3** and **8** were those that contributed the most to the control of lipid oxidation in the SEO. In contrast, the presence of compound **5** negatively affected the behavior of the extracts.

## 3. Material and Methods

### 3.1. Chemicals and Reagents

For SFE, CO_2_ (purity 99.9%, *v*/*v*) and EtOH 96% (*v*/*v*) were purchased from Linde S.A. (Bogotá, Colombia) and from Empresa de Licores de Cundinamarca (Bogotá, Colombia), respectively. Folin–Ciocalteu reagent and pure grade gallic acid were obtained from Merck (Darmstadt, Germany); anhydrous sodium carbonate was from JT Baker; 2,2 diphenyl-1-picrylhydrazyl (DPPH), Trolox, quercetin, and ellagic acid were from Sigma Aldrich Corp (St. Louis, MO, USA); absolute ethanol was from Panreac (Barcelona, Spain); and the tertiary butylhydroquinone (TBHQ) was from TCI (Tokyo, Japan). Solvents used for the UHPLC-DAD were of HPLC grade, from J. T. Baker (Ecatepec, México). Ultrapure water was produced using a Milli-Q system (Billerica, MA, USA).

### 3.2. Oil Sample

The refined, bleached, and deodorized sunflower edible oil (SEO), without antioxidants, was supplied by Team Foods (Bogotá, Colombia).

### 3.3. Sample Treatment

Mango (*Mangifera indica* L.) seeds *var*. Sugar were obtained from a batch of a local agro-industrial processor (Pulpas Oni S.A.S., Bogotá, Colombia). The residual pulp was manually separated and the kernel was separated from the endocarp. Two kilograms of mango seed kernel (MSK) were vacuum-dried (150 mbar) in an oven (VWR Scientific 1410 Vacuum Oven, Cornelius, OR, USA), ground using a grain mill (Corona-Universal, Bogotá, Colombia), and sieved to a particle size lower than 0.30 mm. The ground MSK was stored at room temperature in dark place until use.

### 3.4. Soxhlet Extraction

For the purpose of comparing SFE with a conventional extraction method, Soxhlet (SX) extraction was used. A solute to solvent ratio of 1.0 g: 25.0 mL, with 96% (*v*/*v*) EtOH by reflux of 8 h was used. To reduce the temperature of extraction (T = 45 °C) and therefore to avoid degradation of thermosensitive analytes, a vacuum system (150 mbar) was used. After SX extraction was completed, the lipid fraction was separated from the whole extract, by means of a winterization process. The whole extract was frozen and centrifuged at −20 °C. (Hettich Universal 320R, Tuttlingen, Baden-Württemberg, Germany). Immediately after, the two immiscible phases were filtrated and two fractions were obtained: The solid retained on the filter, called non-soluble fraction (*uSF*), and the filtrated liquid, referred to as soluble fraction (*SF*). The remaining solvent in *uSF* and *SF* was removed using a rotary evaporator (Buchi Rotavapor^®^ R-300, Flawil, Switzerland). The yields in each fraction (*uSF* and *SF*), as well as the overall yield (adding up the uSF and the SF), were evaluated by a gravimetric method. The *SF* under SX extraction was stored in dark place at −20 °C until use. This fraction was evaluated as a potential inhibitor of lipid oxidation when added to an SEO. Moreover, it was characterized measuring the total phenolic content (TPC), total flavonoid content (TFC), and the radical scavenging activity (DPPH assay). SX extraction was performed in triplicate.

### 3.5. Extraction with Supercritical CO_2_ and EtOH as Co-Solvent

A schematic diagram of the equipment used for SFE is shown in Appendix A. Extractions were performed in a home-built experimental unit (at the High-Pressure Laboratory, Universidad Nacional de Colombia, Colombia). The equipment used comprised a 5 mL extraction cell (316L), covered with an electric heat jacket connected to a PID temperature controller. CO_2_ was pressurized through a diaphragm pump (Nova-Swiss 554-2121, Effretikon, Switzerland), connected to a frequency regulator. EtOH (96%, *v*/*v*) was pumped according to the experimental design by using an HPLC pump (Beckman 140A model, Indianapolis, IN, USA). The connection pipes were of 316 stainless steel with a diameter of 1/8”. Each extraction was performed for 3 h at a flow of 10 g CO_2_ min^−1^. The system for the separation of CO_2_ and extract consisted of a back-pressure regulator (BPR, Pressure Tech LF540, Houston, TX, USA) and a separator, in which a depressurization was carried out at 5 MPa and 50 °C to avoid the formation of dry ice due to a high-pressure drop. Finally, the extract was accumulated in a vessel collector at laboratory conditions. For the SFE strategy, the extraction cell was filled with approximately 5.0 g of dry sample and experimental conditions were set for the extraction. Extraction was carried out in a semi-continuous mode. Similar to the Soxhlet extraction, two fractions were obtained and separated (*uSF* and *SF*). Once each fraction was obtained and yield measured, the *SF* was stored in a dark place at –20 °C until use.

A factorial central composite experimental design 2^3^ (with five replicates in the central point) was undertaken to observe the effect of (a) extraction pressure (11.0–21.0 MPa), (b) temperature (40.0–60.0 °C), and (c) composition of EtOH (96% *v*/*v*) in the solvent mixture (5.0–15.0%) on the induction time (t_i_)—the response variable—when the obtained *SF* extracts were added to a bulk oil system. Likewise, the yield (yield of the *SF*, *uSF*, and the overall yield) and TPC, TFC, and DPPH of the *SF* were evaluated (by triplicate) to explain the behavior of the induction time. The experimental conditions are shown in a randomized order in Table 1.

### 3.6. Inhibition of Lipid Oxidation of Sunflower Oil by the Rancimat Method

An accelerated oxidation test using an antioxidant-free SEO was assayed by the Rancimat method. Three grams of SEO added with extract or TBHQ were taken into a reaction vessel. The *SF* of extracts were tested at a concentration of 1000 mg Kg^−1^ at an airflow of 20 L h^−1^ and 120 °C. The t_i_ of the extracts was compared with the one obtained with a control (SEO without antioxidants). For the sake of comparison, the t_i_ with TBHQ (200 mg Kg^−1^) was also tested. Concentration of TBHQ was selected according to the Alimentary Codex [45,46]. The results were expressed as t_i_ and protection factor (PF), where t_i antioxidant_ and t_i control_ were induction time for SEO with extract or TBHQ and without extract, respectively (Equation (1)).
(1)$$\mathrm{PF}=\frac{t_{{i\;}_{\mathrm{antioxidant}}}}{t_{{i\;}_{\;\mathrm{control}}}}$$

### 3.7. Total Phenolic Content

The total phenolic content (TPC) in the obtained extracts was measured, using the Folin–Ciocalteu reagent and followed with some modifications to the methodology reported by Carrillo et al. [47]. For the determination, 100 μL of extract solution (10 mg mL^−1^) was mixed with 100 μL Folin–Ciocalteu reagent, shaken and left to stand for 5 min in a dark place. Then, 300 μL of aqueous Na_2_CO_3_ (20%, *w/v*) was added, shaken, and left to stand for 90 min in a dark place while it reacted. The absorbance of the blue-colored solution was measured as 765 nm in a UV-Vis spectrophotometer (Thermo-Scientific Genesys 10 UV-Vis, Waltham, MA, USA). Gallic acid (GA) was used to build a standard curve (20–140 mg GA L^−1^, R^2^ = 99.71%) for the determination. The TPC was expressed in milligrams of GA equivalents per g of extract (mg-eq GA g^−1^ extract).

### 3.8. Total Flavonoid Content

Total flavonoid content (TFC) was determined using the aluminum chloride assay through colorimetry, as described by Chang et al. [48]. Each extract solution (500 µL at 20 mg mL^−1^) was mixed with 1500 µL of 100% *v*/*v* EtOH, 100 µL of 10% *w/w* aluminum chloride, 100 µL of 1 M sodium acetate, and 2800 µL of water. The mixture was stirred and incubated at room temperature for 30 min. After such reaction time, the absorbance was measured at 415 nm in a spectrophotometer Thermo-Scientific Genesys 10 UV-Vis (Waltham, MA, USA). The TFC results were expressed as mg of quercetin (Q) equivalents per g of extract (mg-eq Q g^−1^ extract), by using a Q calibration line (20–160 mg Q L^−1^, R^2^ = 99.65%).

### 3.9. DPPH Radical Scavenging Antioxidant Activity

Radical scavenging activity of the extracts was evaluated by the DPPH assay, following with certain modifications the procedure described by Carrillo et al. [47]. The test was carried out using 950 μL of a 100 µM DPPH solution prepared in EtOH, and it was mixed with 50 μL of either extract (at 10 mg extract mL^−1^) or standard (Trolox) in EtOH. As a negative control, the same amount of DPPH and 50 μL of EtOH were used. The mixtures were shaken and left to react at room temperature in a dark place for 60 min. The absorbance was registered at 517 nm in a spectrophotometer (Thermo-Scientific Genesys 10 UV-Vis, Waltham, MA, USA). The results were expressed as a percentage of inhibition of DPPH radical (Equation (2)) and as TEAC values (µmol Trolox g^−1^ extract). TEAC values were calculated by means of a calibration curve using different concentrations of the standard Trolox (125–3000 µM, R^2^ = 99.15%).
(2)$$\%Inhibition=\frac{\left(A_{o}-A_{f}\right)}{A_{o}}\times 100$$
where *A_o_* refers to the absorbance at t = 0 and *A_f_* refers to the absorbance after 30 min of reaction.

### 3.10. Evaluation of Phenolic Compounds by RP-UHPLC-DAD

Extracts obtained by the SX and SFE methods as described in Section 2.4 and Section 2.5 were analyzed by RP-UHPLC-DAD. Chromatographic conditions were those described by Cuellar et al. [49]. Extracts were injected at a concentration of 1.0 mg L^−1^. Authentic standards of gallic acid, ellagic acid, mangiferin, (+)-catechin, and (–)-epicatechin were injected under the same conditions the extracts were analyzed. Tentative identification was based on the comparison of retention time and UV-Vis spectra against the authentic standards. The compounds were quantified by the external standard method. Gallic and ellagic acids were quantified with authentic standards (six data points, ranging from 1.0 to 100.0 mg L^−1^, R^2^ = 99.80 and 99.91% for gallic and ellagic acids, respectively), other compounds were quantified as ellagic acid equivalents. The results were expressed as mg either gallic or ellagic acid g^−1^ extract. No correction for different molecular weights or for different molar extinction coefficients was applied to the compounds different from gallic acid or ellagic acid.

### 3.11. Statistical Analysis

Values for t_i_, TPC, TFC, DPPH, and the concentration of each individual phenolic compound were expressed as an average ± SD (standard deviation), with three replicates. Normality of the dataset was tested by the Kolmogorov–Smirnov test. Correlation among the response variables was evaluated by Pearson’s correlation coefficient (*p* < 0.05). For the results of SFE assays, an analysis of variance (ANOVA) was carried out (*p* < 0.05). A second-degree mathematical model was tested to evaluate how the t_i_ was correlated with the factors pressure, temperature, and %EtOH. The lack of fit and the adjusted coefficient of determination (R^2^) were calculated to assess the adequacy of the mathematical model. Stepwise backward elimination was used to eliminate those regression coefficients without statistical significance. Those with statistical significance (*p* < 0.05) were kept. The mathematical model in the SFE was represented by a surface response.

Linear and quadratic mathematical models were tested to evaluate if the TPC, TFC, and DPPH were responsible for the t_i_ when each extract was used to protect an edible oil. Similarly, linear and quadratic models were used to evaluate if the concentration of the individual phenolics in the extracts was responsible for the observed t_i_. The adjusted R^2^ and the absolute average deviation were calculated to estimate the adequacy of each mathematical model under study. Stepwise backward elimination was used to eliminate those regression coefficients without statistical significance; those with statistical significance were kept (*p* < 0.05).

All statistical analyses were performed with Statgraphics^®^ Centurion XVI software (Manugistics, Inc., Rockville, MD, USA).

## 4. Conclusions

Supercritical fluid extraction (SFE) was found to be a suitable extraction method to obtain extracts from the mango seed kernel (MSK), which showed antioxidant activity and modified the oxidative stability of antioxidant-free sunflower edible oil (SEO). SC-CO_2_ + EtOH allowed for obtaining extraction yields, TPC and TFC equal to or greater than those obtained by the Soxhlet (SX) extraction. The extract that showed the highest protection against oxidation of SEO (t_i_ = 6.1 h) was obtained at 21.0 MPa, 60.0 °C and 15.0% EtOH, with a total yield of 11.8%, TPC 19.4 mg-eq AG g^−1^ extract, TFC 3.8 mg-eq Q g^−1^ extract, and percentage of inhibition of DPPH radical of 36.0%. Gallic acid and ellagic acid were found in the extract. The composition of the extract was found to influence the stability of the SEO. Finally, it can be concluded that it is possible to obtain extracts from Colombian MSK *var*. Sugar with antioxidant activity comparable to that of commercial antioxidants (TBHQ).

The set of results obtained provides the basis for proposing the use of mango seed *var*. Sugar—an important agro-industrial by-product in the fats and oils industry. The MSK could be a source of antioxidant extracts applied as additives in edible oils. This work is framed within the guidelines of green chemistry and it was proposed by other authors as a feasible way to deal with large amounts of agro-industrial residues. Lavecchia and Zuorro obtained flavonoid-rich extracts from the olive pomace by a classic extraction technique [50]; Lima et al. obtained promissory ingredients for functional foods by processing guava waste under ultrasound extraction [51]; Dominguez et al. proposed further research into some Passiflora species peels so as to obtain antioxidant extracts rich in phenols under pressurized liquid extraction [52]. Indeed, our work and that of other authors are contributing to new research topics in Green Foodomics [53].

## Figures and Tables

**Figure 1 molecules-26-07516-f001:**
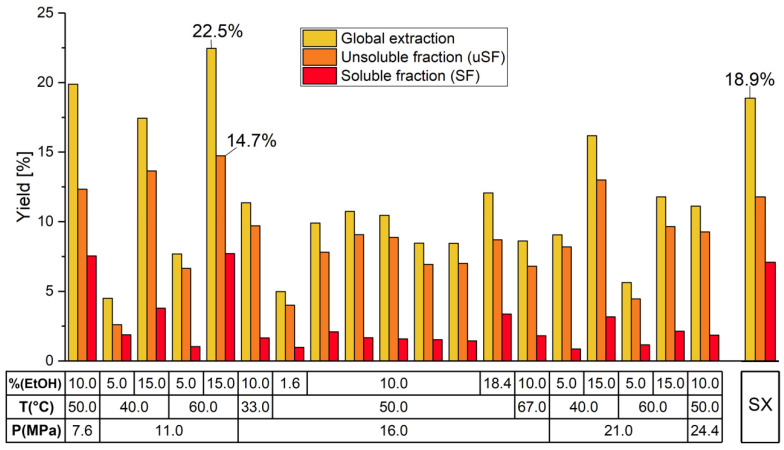
Extraction yield under the supercritical CO_2_ + EtOH and Soxhlet (SX) extraction methods.

**Figure 2 molecules-26-07516-f002:**
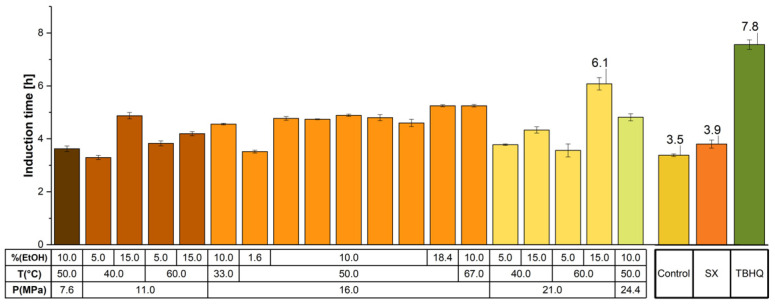
Oxidative stability of SEO as measured by the induction time (h) using the Rancimat method. SEO was added with SFE and Soxhlet extracts, and TBHQ as synthetic antioxidant, an SEO without antioxidants was included (Control).

**Figure 3 molecules-26-07516-f003:**
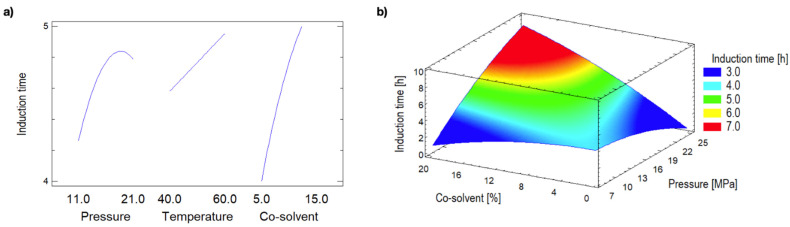
Main effects (**a**) and response surface predicted for the induction time of sunflower oil added with extracts obtained by supercritical fluid extraction at 67.0 °C (**b**).

**Figure 4 molecules-26-07516-f004:**
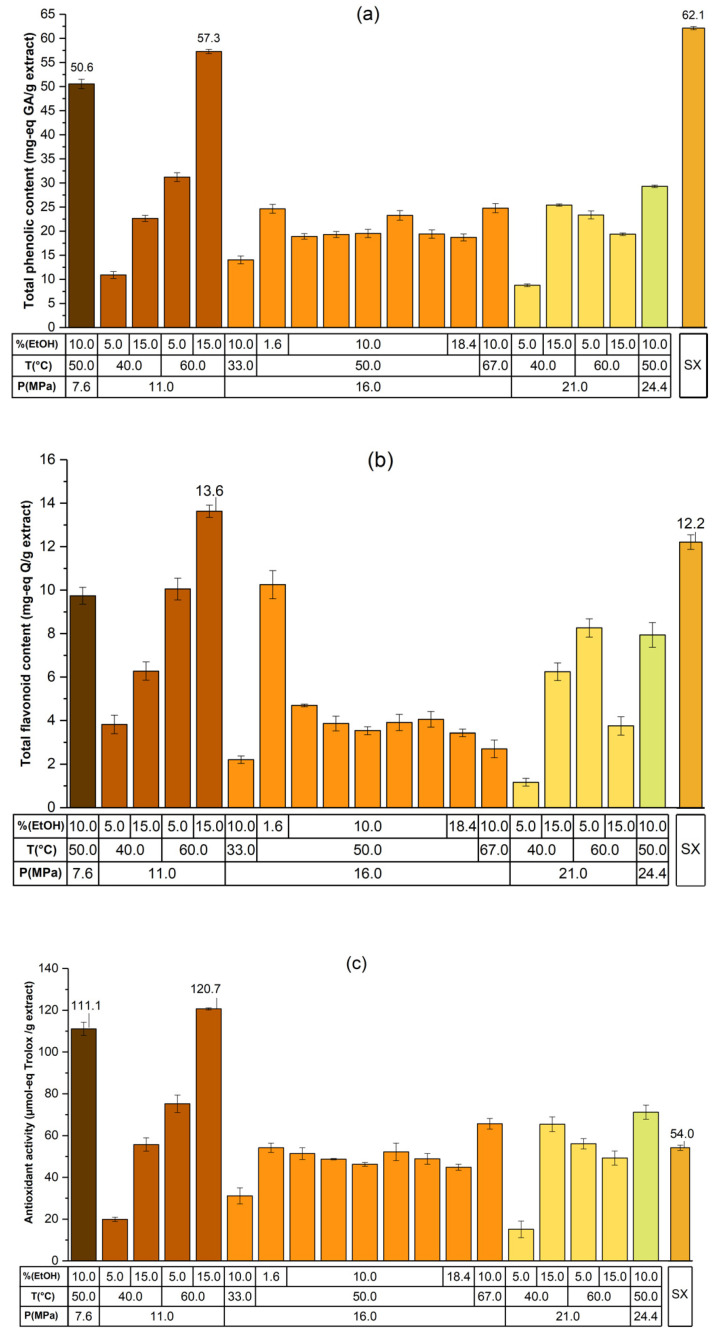
Total phenolics content (**a**), total flavonoids content (**b**), and antioxidant activity (**c**) as measured by the radical scavenging method from supercritical fluid and Soxhlet extracts.

**Figure 5 molecules-26-07516-f005:**
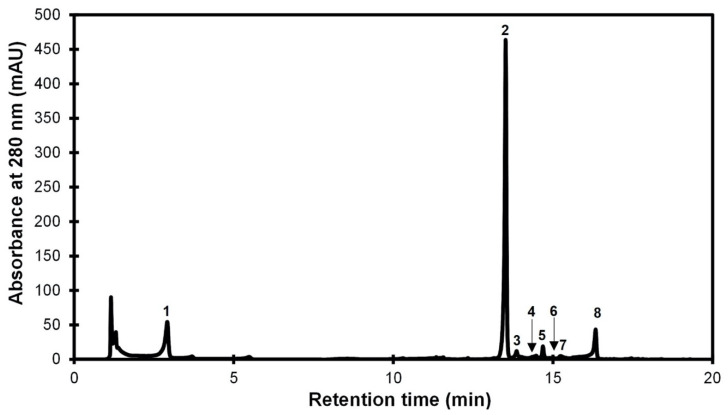
RP-UHPLC analysis of the extract obtained by supercritical fluid extraction at 21.0 MPa, 60.0 °C, and 15.0% EtOH.

**Table 1 molecules-26-07516-t001:** Central composite experimental design for the preparation of extracts from mango seed kernel employing supercritical fluid extraction (CO_2_-EtOH).

Experiment	Pressure	Temperature	Co-solvent
#	(MPa)	(°C)	(%)
**1**	11.0	60.0	15.0
**2**	16.0	50.0	10.0
**3**	7.6	50.0	10.0
**4**	24.4	50.0	10.0
**5**	16.0	50.0	18.4
**6**	16.0	33.0	10.0
**7**	21.0	40.0	5.0
**8**	16.0	50.0	10.0
**9**	11.0	40.0	5.0
**10**	21.0	60.0	15.0
**11**	11.0	40.0	15.0
**12**	21.0	60.0	5.0
**13**	16.0	50.0	10.0
**14**	16.0	67.0	10.0
**15**	16.0	50.0	10.0
**16**	11.0	60.0	5.0
**17**	21.0	40.0	15.0
**18**	16.0	50.0	1.6
**19**	16.0	50.0	10.0

Experiments are presented as they were performed, in a randomized order. Underlined values represent the central points of experimental design.

**Table 2 molecules-26-07516-t002:** Regression coefficients for the induction time (t_i_).

Coefficient	Estimate	*p*-Value
a	−8.8966	
b	0.8911	0.0008
c	0.1879	0.0028
d	1.3504	0.0000
e	−0.0097	0.0010
f	−0.0117	0.0050
g	−0.0737	0.0195
h	−0.0235	0.0664
i	−0.0074	0.0028
j	0.0016	0.0004
Lack of fit	0.0596	
Adjusted R^2^	92.4%	

Coefficients are given for the equation *t*[*h*] = a + b**P** +c**T** + d**C** + e**P^2^** + f**PT** + g**PC** + h**TC** + i**C^2^** + j**PTC**, where **P** is pressure, **T** is temperature, and **C** is %EtOH.

**Table 3 molecules-26-07516-t003:** Linear mathematical model, after stepwise backward elimination, to evaluate how the composition of the extract can explain the observed induction time.

Parameter	Estimate	Standard Error	T Statistic	*p*-Value
Constant	2.9324	0.2885	10.1628	0.0000
Peak 1	0.0315	0.0066	4.7662	0.0002
Peak 3	0.0342	0.0104	3.3071	0.0048
Peak 5	−0.0146	0.0037	3.9829	0.0012
Peak 8	0.0328	0.0119	2.7599	0.0146

Adjusted R^2^: 89.3%. Absolute standard deviation: 4.6%.

## Data Availability

Data are presented in the paper and available from the authors.

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
