# Peer review of "Supercritical Fluid Extraction of Phenolic Compounds from Mango (Mangifera indica L.) Seed Kernels and Their Application as an Antioxidant in an Edible Oil"

_molecules, 2021, doi:10.3390/molecules26247516_

Round 1

Reviewer 1 Report

After evaluating the manuscript "Green extraction of phenolic compounds from mango (Mangifera indica L.) seed kernel and their application as antioxidant in an edible oil" I have to recommend its major revision in the current version.  Major 1. All figures need to be improved (style and resolution in the first place) 2. Authors should pay more attention to the composition of the isolated product. 3. Alternative methods for testing antioxidant activity should be applied (for example 10.1016/j.jsps.2012.05.002)

Author Response

The correction of the reviewers was much appreciated, and we revised our manuscript accordingly. All suggestions from the reviewers were considered.

Revision:

I have to recommend its major revision in the current version. 

Answer: The new manuscript version shows the SFE experimental conditions at the first table. We believe that this change let see with more clarity the results and discussion. Better information was provided at Introduction. New abbreviations were used and defined. English was improved.

  1. All figures need to be improved (style and resolution in the first place).

Answer: All figures were improved, it let have a more clarify manuscript.

  1. Authors should pay more attention to the composition of the isolated product.

Answer: UHPLC results were additioned as supplementary material (about extract components and used standards).

  1. Alternative methods for testing antioxidant activity should be applied (for example 10.1016/j.jsps.2012.05.002).

Answer: We have used DPPH method because it is the most used to measured antioxidant activity. The main important studied topic is development and obtain new biomaterial and its extracts with the property of to transfer oxidative stability-OS to edible oils. Rancimat method is the most used method to study OS.

Reviewer 2 Report

The quality of the figures must be improved, it is a bit difficult to interpret them

Although SFE is part of the Green Extraction technologies, it is not all, so perhaps in the title that word should be replaced by SFE

There are investigations similar to the one proposed here
What  is the real contribution?

Although SFE is part of the Green Extraction technologies, it is not all, so perhaps in the title that word should be replaced by SFE

There are investigations similar to the one proposed here
What as authors is your contribution?

Minors:

Why was only DPPH used to evaluate antioxidant activity?

Line 160 is confusing, please correct

In the methodology, please describe how the mango kernel was dried and ground

The quality of the figures must be improved, it is a bit difficult to interpret them

Author Response

The correction of the reviewers was much appreciated, and we revised our manuscript accordingly. All suggestions from the reviewers were considered.     Revision:
  1. The quality of the figures must be improved, it is a bit difficult to interpret them.

Answer: All figures were improved, it let have a more clarify manuscript.

  1. Although SFE is part of the Green Extraction technologies, it is not all, so perhaps in the title that word should be replaced by SFE.

Answer: Title was modified.

  1. There are investigations similar to the one proposed here. What is the real contribution? What as authors is your contribution?

Answer: The real and main important contribution of this work it was obtained a promissory extract that give oxidative stability to edible oils from an agro-industrial by product. It was obtained from mango seed kernel, which is consistent with green chemistry principles.

  1. Why was only DPPH used to evaluate antioxidant activity?

Answer: We have used DPPH method because it is the most used to measured antioxidant activity. The main important studied topic is development and obtain new biomaterial and its extracts with the property of to transfer oxidative stability-OS to edible oils. Rancimat method is the most used method to study OS.

  1. Line 160 is confusing, please correct.

Answer: The text was corrected.

  1. In the methodology, please describe how the mango kernel was dried and ground.

Answer: Methodology was completed.

Reviewer 3 Report

Prof. Fabian’s group has shown us an interested and detailed research which will reduce the waste from agricultural productions.

It’s a comfort to read through a well-organized manuscript. All the information strongly supports the importance of this extract.

This manuscript will be accepted and published in this journal after studying the addition of third organic solvent to sure the polarity effect.

Author Response

The correction of the reviewers was much appreciated, and we revised our manuscript accordingly. All suggestions from the reviewers were considered.   Revision:  

This manuscript will be accepted and published in this journal after studying the addition of third organic solvent to sure the polarity effect.

Answer: In accordance with previous results of our group, in mango seeds and other seeds (guava or papaya seeds) EtOH was used as a cosolvent. When EtOAc was used as a cosolvent, the extracts obtained did not have greater antioxidant activity than those obtained with EtOH as a cosolvent. In addition, the polarity in supercritical fluid conditions is a function of density, which depends of pressure and temperature.

Round 2

Reviewer 1 Report

The manuscript by Fabián Parada-Alfonso et al is substantially improved. The authors have adequately addressed all of my concerns, and (IMHO) the concerns of the other reviewers as well.. I support immediate publication of the manuscript.

Author Response

Dear reviewer,
We really appreciate the time dedicated to review our research work. Your comments and the those others helped us a better document have. English language has been revised.